# Puerarin Reduces Oxidative Damage and Photoaging Caused by UVA Radiation in Human Fibroblasts by Regulating Nrf2 and MAPK Signaling Pathways

**DOI:** 10.3390/nu14224724

**Published:** 2022-11-09

**Authors:** Qiuting Mo, Shuping Li, Shiquan You, Dongdong Wang, Jiachan Zhang, Meng Li, Changtao Wang

**Affiliations:** 1Beijing Advanced Innovation Center for Food Nutrition and Human Health, Beijing Technology and Business University, Fucheng Road, Beijing 100048, China; 2Institute of Cosmetic Regulatory Science, College of Chemistry and Materials Engineering, Beijing Technology & Business University, Fucheng Road, Haidian District, Beijing 100048, China; 3Beijing Key Laboratory of Plant Resource Research and Development, College of Chemistry and Materials Engineering, Beijing Technology and Business University, Fucheng Road, Haidian District, Beijing 100048, China

**Keywords:** puerarin, oxidative stress, antioxidant, anti-photoaging

## Abstract

Fibroblasts account for more than 95% of dermal cells maintaining dermal structure and function. However, UVA penetrates the dermis and causes oxidative stress that damages the dermis and accelerates skin aging. Puerarin, the main active ingredient of *Puerariae lobata*, has been demonstrated to withstand oxidative stress caused by a variety of factors. However, there are limited findings on whether puerarin protects fibroblasts from UVA-induced oxidative stress damage. The effects of puerarin on human skin fibroblasts (HSF) under UVA-induced oxidative stress were investigated in this study. It is found that puerarin upregulates antioxidant enzymes’ mRNA expression level and their content through modulating the KEAP1-Nrf2/ARE signaling pathway, thus improving cell antioxidant capacity and successfully eliminating UVA-induced reactive oxygen species (ROS) and lipid oxidation product malondialdehyde (MDA). Additionally, puerarin blocks the overexpression of human extracellular signal-regulated kinase (ERK), human c-Jun amino-terminal kinase (JNK), and P38, which downregulates matrix metalloproteinase 1 (MMP-1) expression and increases type I collagen (COL-1) expression. Moreover, preliminary research on mouse skin suggests that puerarin can hydrate, moisturize, and increase the antioxidant capacity of skin tissue. These findings suggest that puerarin can protect the skin against photoaging.

## 1. Introduction

Ultraviolet light is a representative substance that causes oxidative stress in daily life. UVA (320–400 nm) is the most exposed ultraviolet radiation with the highest total radiation intensity in daily life, and it causes skin damage, induces skin aging, and weakens the protective effects of the skin by damaging the DNA, protein, and other biological macromolecules [1]. In the 3D skin model, the skin surface became rougher, the structures of the epidermis and dermis were destroyed, and the DNA chain was broken after UVA irradiation [2].

The KEAP1-Nrf2/ARE signaling pathway is closely related to oxidative stress. Nuclear factor-erythroid2-relatedfactor2 (Nrf2), a key transcription factor for regulating antioxidant stress, exists in various body organs and stably couples with the cytoplasmic protein Kelch-like chloropropane-related protein-1 (KEAP1) in the cytoplasm in normal physiological environments until oxidative stimulators attack the cells; it dissociates from KEAP1 and is transferred to the nucleus where it binds to antioxidant response elements (ARE) and activates the expression of downstream antioxidation-related factors, such as heme oxygenase-1 (HO-1), human quinone NADH dehydrogenase 1 (NQO1), Catalase (CAT), Superoxide Dismutase (SOD), and the L-γ-glutamyl-cysteinyl glycine redox system, playing an essential role in inducing antioxidant response [3,4]. Therefore, studying the activators of the KEAP1-Nrf2-ARE pathway is critical for preventing and treating illnesses caused by oxidative stress.

*Puerariae lobata* is a traditional Chinese medicinal and edible plant attracting increasing attention by virtue of its application value in pharmacology and health care and its advantages such as low cost, low toxicity, and few side effects [5,6]. Numerous experimental studies have proven that *Puerariae lobata* not only resists oxidative stress, protects nerve tissue, dispels the effects of alcohol, allays fever, reduces blood sugar level, and lowers blood pressure, but also is used to treat arteriosclerosis, angina pectoris, coronary atherosclerotic heart disease, and tumors [7]. Puerarin, one of the most important active components, is a kind of isoflavone of *Puerariae lobata* [8]. Studies have shown that puerarin has a strong antioxidant capacity, enabling it to significantly improve the activity of antioxidant enzymes, and reduce the content of oxidative products, showing a defense against oxidative stress in various disease models [9,10] and cell damage models [11,12]. Moreover, it reduces renal injury induced by carbon tetrachloride [13], traumatic brain injury caused by bench-mark TM [14], and sodium pentobarbital-induced heart failure [15] by inhibiting oxidative stress via the modulation of the KEAP1-Nrf2/ARE pathway.

However, there are few reports on whether puerarin has protective effects on oxidation-damaged skin fibroblasts caused by UVA. Thus, we sought to determine if pretreatment with puerarin can help skin resist oxidative stress through human skin fibroblasts (HSF) and Kunming mouse oxidative stress models.

## 2. Materials and Methods

### 2.1. Chemicals

Human Skin Fibroblasts (HSF) were provided by the Chinese Academy of Sciences. Puerarin was purchased from Shanghai Yuan Ye Bio-Technology Co., Ltd., China (Shanghai, China). Fetal bovine serum, penicillin–streptomycin, Dulbecco’s Modified Eagle’s Medium (DMEM), and trypsin were purchased from Gibco, USA (Shanghai, China). Cell Counting Kit-8 (CCK-8), Reactive Oxygen Species Assay Kit, Lipid Peroxidation MDA Assay Kit, Total Antioxidant Capacity Assay Kit with ABTS method, Total Superoxide Dismutase Assay Kit with WST-8, Cellular Glutathione Peroxidase Assay Kit with NADPH, Catalase Assay Kit, BCA Protein Assay Kit, Trizol, nuclear and cytoplasmic protein extraction kit and cell lysis buffer were purchased from Beyotime (Shanghai, China). Human Kelch-like ECH-associated protein 1(KEAP1) ELISA, Human Nuclear Factor E2-related Factor 2 (Nrf2) ELISA, Human Quinone NADH Dehydrogenase 1 (NQO1) ELISA, Human Heme Oxygenase-1 (HO-1) ELISA, Human GCLC (Glutamate-cysteine ligase catalytic subunit) ELISA and Human Matrix metalloproteinase 1, MMP-1 ELISA kits were purchased from Cusabio Biotech Co., Ltd. (Houston, TX, USA). ELISA Kit for Collagen Type I (COL-1) was purchased from Cloud-Clone Corp, USA (Wuhan, China). TransStart^®^ Top Green qPCR SuperMix Kit and EasyScript^®^ All-in-One First-Strand cDNA Synthesis SuperMix for qPCR (One-Step gDNA Removal) Kit were purchased from Beijing TransGen Gold Biotech Co., Ltd., China (Beijing, China).

### 2.2. Cell Culture

HSF were grown in a DMEM medium supplemented with 10% fetal bovine serum and 1% Penicillin–Streptomycin Solution. The incubator was humidified with a temperature of 37 °C and CO_2_ content of 5%. Trypsin was used for digestion and passage when the cell density was around 85%.

### 2.3. Measurement of Cytotoxicity

HSF in good condition at the logarithmic growth stage were seeded on 96-well plates at a density of 8 × 10^3^ cells/well for 12 h, then puerarin solution was added at concentrations of 0, 6.25, 12.5, 25, 50, 100, 200, and 400 µg/mL. Puerarin was pre-dissolved with dimethyl sulfoxide (DMSO), followed by gradient dilution with DMEM without fetal bovine serum. The concentration of DMSO in 400 µg/mL puerarin solution was 0.1%. After puerarin treatment for 24 h [16], it was changed to 100 µL serum-free DMEM containing 10% CCK8 solution for 2 h. The CCK-8 approach is based on 2-(2-Methoxy-4-nitrophenyl)-3-(4-nitrophenyl)-5-(2,4-disulfophenyl)-2H-tetrazoliumsodiumsalt (WST-8) which is an upgraded alternative to 3-(4,5)-dimethylthiahiazo (-z-y1)-3,5-di-phenytetrazoliumromide (MTT). Absorbance was determined at 450 nm.

### 2.4. UVA Irradiation Procedure

HSF were seeded for 12 h until they adhered to the plate. After gentle washing with PBS, HSF were placed in a UVA box for radiation at intensities of 0, 3, 6, 9, and 12 J/cm^2^. The IC 50 value of the cell survival rate calculated by SPSS was about 12 J/cm^2^, so HSF treated with 12 J/cm^2^ UVA was used as a model of oxidative damage.

### 2.5. Protective Effects of Puerarin on UVA-Induced Oxidative Stress

The protective effects of puerarin on oxidative stress were evaluated by detecting whether puerarin improves the viability of oxidation-stimulated cells. After successful cell adherence, puerarin solution was applied for 24 h, followed by 12 J/cm^2^ UVA irradiation. Cell viability was determined using the CCK-8 method.

### 2.6. Intracellular ROS Content Measurement

After puerarin treatment for 24 h, HSF were incubated with 100 µL 20 µM 2,7-dichlorodihydrofluorescein diacetate (DCFH-DA) for 40 min to detect ROS production. The excess DCFH-DA was then washed away with PBS before HSF were exposed to UVA radiation. Subsequently, HSF were detected (excitation wavelength of 488 nm, emission wavelength of 525 nm).

### 2.7. Measurement of Intracellular Total Antioxidant Capacity (ABTS Method), MDA Content, and Antioxidant Enzyme Activity

HSF were placed in 6-well plates and treated with puerarin, followed by UVA stimulation. The treated HSF were placed in a low-temperature environment, and a 100 µL cell lysis buffer was added. Finally, the supernatant was centrifuged at 4 °C for 10 min (10,000 r/min) and stored at −80 °C for subsequent antioxidant determination. According to the kit’s instructions, total antioxidant capacity, MDA content, and antioxidant enzyme activity were measured.

### 2.8. ELISA

The cells were plated in 25 cm^2^ flask at density of 5 × 10^6^ cells/flask or 75 cm^2^ flask at density of 1.5 × 10^7^ cells/flask for 12 h. After sample processing and UVA irradiation, the nuclear and cytoplasmic proteins were prepared according to nuclear and cytoplasmic protein extraction kit. Then, the protein of Nrf2 in nuclear and cytoplasmic was detected. The cell lysates were prepared by cell lysis buffer. Total protein was estimated from the cell lysates for KEAP1, NQO1, HO-1, GCLC, and COL-1 assay. Cell Culture Supernates were used to determine the protein content of MMP-1.

For details about test protocols, refer to the kits’ instructions.

### 2.9. RT-PCR

To extract RNA from the treated HSF, the Trizol reagent was used, followed by reverse transcription. Referred to the instructions for specific operations. The primers were designed using Primer Express^TM^ software v3.0.1 based on the gene sequences published by National Center for Biotechnology Information (NCBI), and the reference gene was β-actin. The mRNA expression level of the cells was determined using reverse transcription polymerase chain reaction (RT-PCR). The cycle parameters were 94 °C for 30 s, then 94 °C for 15 s, 60 °C for 15 s, and 72 °C for 10 s, a total of 40 cycles, and the fluorescence data were collected at 72 °C. The reaction was performed using the QuanStudio3 fluorescence quantitative PCR instrument. Table 1. Shows the RT-PCR primer sequences:

### 2.10. Experiments on Animals

#### 2.10.1. Experimental Animals and Groups

Twenty female mice weighing 18 ± 2 g were purchased from Beijing Vital River Laboratory Animal Technology Co., Ltd (Beijing, China). Then they were divided into 4 groups randomly: blank group (deionized water), model group (deionized water + UVA), positive control group (50 μg/mL VC + UVA), and puerarin group (12.5 μg/mL + UVA). Puerarin was pre-dissolved with DMSO and then diluted with distilled water. The concentration of DMSO in 12.5 μg/mL puerarin solution was less than 0.1%. Each group was housed in a cage with a temperature of 22–25 °C and a circadian cycle of 12 h of light and 12 h of darkness. They consumed food and liquids ad libitum. After one week of acclimation to the environment, the experiment was carried out. The same part of the back of each mouse was depilated with depilation cream, and the depilation area was 3 × 3 cm. After 1 h, the sample was applied at the dosage of 1 mL/d. After 30 min, they were irradiated with UVA at a dose of 50 J/cm^2^. The experimental period was 30 days [17].

#### 2.10.2. Moisturizing Experiment

After two months of feeding and before the mice were euthanized, the water content and transepidermal water loss of the back skin were measured using a water content tester (Corneometer^®^ CM825, Courage + Khazaka Co., Köln, Germany) and transepidermal water loss tester (Tewameter^®^ TM300, Courage + Khazaka Co., Köln, Germany) (Temperature: 27 ± 1 °C, humidity: 40 ± 5%).

#### 2.10.3. Skin Tissue Total Antioxidant Capacity and CAT Activity Measurement

Skin tissues removed from the same part of the mice’s back were cut into small fragments (100–150 mg), and 1 mL of cell lysis buffer was added and homogenized with a glass homogenizer until the skin tissue was completely lysed. After centrifugation for 5 min (12,000× *g*), the CAT activity and total antioxidant capacity of the supernatant were determined.

#### 2.10.4. Hematoxylin-Eosin (HE) Dye

Other skin tissues were immersed in a 4% formaldehyde solution, dehydrated by ethanol gradient, and embedded in paraffin, then the wax block was continuously cut into 4 μm thin slices, dewaxed, and conventionally rehydrated. The skin was then soaked in hematoxylin for 5 min and rinsed under tap water, then differentiated in 1% alcohol hydrochloride for 5 s and rinsed in tap water for 1 h. After saturation in a lithium carbonate aqueous solution for 2 min, the samples were washed in tap water for 30 min. Finally, they were soaked in a 0.5% eosin solution for 2 min and sealed with neutral gum. The structural characteristics of the skin tissues in each group were observed under a microscope.

### 2.11. Statistical Analysis

Each experiment included at least three biological repeats with three technical replicates. All variables were reported as mean ± standard deviation and were processed using IBM’s SPSS v22 software (Armonk, NY, USA). To evaluate statistical significance between groups, a univariate analysis of variance (ANOVA) test was used. *p* < 0.05 was used to determine if the differences were statistically significant.

## 3. Results

### 3.1. Effects of Puerarin on HSF

The cell viability was 97.92% when serum-free DMEM containing 0.1% DMSO was used for the cytotoxicity test, indicating that DMSO had a negligible effect on cells. When the concentration of puerarin solution is 6.25–200 µg/mL, the survival rate of HSF is higher than 80%, and it even shows a proliferation effect on the cells, which is proportional to the concentration. The cell viability rates are 112.67%, 116.92%, 118.29%, 123.55%, 126.51%, and 128.26%, respectively. However, when the concentration is higher than 400µg/mL, the cell viability decreases to 79.39%, which may have certain toxicity to HSF (Figure 1a). Low-dose (3 and 6 J/cm^2^) UVA irradiation show 102% and 83.76% cell viability having no significant effect on cells (Figure 1b). When the UVA irradiation is 9 J/cm^2^, the cell viability is 78.40%, but the effect on cell viability is still relatively small. When the UVA irradiation is 12 J/cm^2^, the cell viability is decreased to 51.74% (*p* < 0.001). Excessive exposure to UVA would cause irreversible damage to cells, so 12 J/cm^2^ irradiation was used to establish the oxidation model. Cell viability is significantly decreased after UVA stimulation (*p* < 0.001) but significantly increase to 80.76%, 86.32%, and 84.43% in order after treatment with 12.5, 25, and 50 µg/mL puerarin solution, indicating protection from oxidative stress by UVA radiation (*p* < 0.001) (Figure 1c). After comprehensive consideration, 12.5 µg/mL of puerarin was finally selected for subsequent experiments.

### 3.2. Effects of Puerarin on Intracellular Antioxidant Capacity

Take 50 µg/mL of vitamin C (VC) as the positive control of the antioxidant test [18]. In the presence of antioxidants, the oxidation of 2,2′-azino-bis (3-ethylbenzthiazoline-6-sulfonic acid) (ABTS), an artificial free radical, to green ABTS+ is blocked [19]. By measuring and calculating the absorbance change, the total antioxidant capacity of the samples may be determined. Compared with the model group (0.903 mmol/g protein), the total antioxidant capacity of the puerarin group and positive control group is improved to 1.46 and 1.39 mmol/g protein, which slowed down the oxidative damage caused by UVA to cells (Figure 2a). Reactive oxygen species (ROS) and lipid oxidation product malondialdehyde (MDA) are the signature products of cellular oxidative stress [20,21]. UVA radiation causes a significant increase in ROS and MDA content (*p* < 0.001) (Figure 2b). However, 12.5 µg/mL of puerarin solution dramatically minimizes the generation of ROS (relative fluorescent value from 2774 to 1167) and MDA (from 4.2 to 1.98 µmol/mg protein) as a result of UVA damage (*p* < 0.001) (Figure 2c).

The effects of puerarin on the KEAP1-Nrf2/ARE signaling pathway in UVA oxidatively damaged cells were investigated in this work from two perspectives: protein content and mRNA expression level. The Puerarin solution reduces the relative mRNA transcription of KEAP1 from 1.743 to 1.302 (Figure 3a) and the protein content from 51.68 to 45.77 ng/mL (Figure 3c). Puerarin promotes the relative mRNA transcription of Nrf2 from 0.353 to 0.448 (Figure 3b). By measuring the relative protein content of Nrf2 in cytoplasm and nucleus, it was found that after UVA treatment, the protein content of Nrf2 in the nucleus decreased to 5.62 ng/mL, and the cytoplasmic protein increased to 15.36 ng/mL, while after puerarin treatment, the nuclear Nrf2 content is reversed to 15.76 ng/mL, and the cytoplasmic Nrf2 content is 7.797 ng/mL. It is hypothesized that puerarin reduces the protein translation of KEAP1 by inhibiting KEAP1 transcription, thereby preventing the degradation of Nrf2 in the cytoplasm. The UVA radiation destroys the intracellular antioxidant system, and the nuclear translocation of Nrf2 is inhibited, leading to the decrease of intracellular Nrf2 protein content. The effect of puerarin restores the cellular antioxidant system and increases the nuclear Nrf2 protein content.

Activated Nrf2 induces the expression of antioxidant enzymes related to cell protection, thereby improving the resistance of cells to oxidative damage. In the experiment of UVA-stimulated HSF, UVA radiation significantly lowers the enzymes activity and mRNA expression of glutathione peroxidase (GSH-Px), CAT, SOD, HO-1, NQO1, and glutamate-cysteine ligase catalytic subunit (GCLC) (*p* < 0.01). However, puerarin has significant promoting effects (Figure 4). The relative transcript levels of antioxidant enzymes reduced by UVA irradiation in the inner cell are enhanced from 0.199 to 1.036, from 0.582 to 1.282, from 0.202 to 3.726, from 0.415 to 0.655, from 0.577 to 2.911, and from 0.597 to 1.741, respectively. The corresponding antioxidant enzyme activity also is increased sequentially from 12,070 to 16,757 units/mg protein, from 6.665 to 10 U/mg protein, from 0.701 to 2.899 mU/mg protein, from 0.409 to 0.67 ng/mL, from 0.256 to 0.292 ng/mL, and from 1.872 to 3.752 ng/mL.

### 3.3. Effects of Puerarin on UVA-Induced Cell Senescence

RT-PCR results show that 12 µg/mL of puerarin solution significantly inhibits the mRNA expression levels of the human c-Jun amino-terminal kinase (JNK) from 1.826 to 0.306, human extracellular signal-regulated kinase (ERK) from 1.526 to 0.454, and P38 from 1.693 to 0.688 upregulated by UVA irradiation compared with the model group (*p* < 0.01) (Figure 5).

UVA-induced oxidative stress upregulates the expression of matrix metalloproteinases (MMPs), leading to skin collagen degradation and other extracellular matrix (ECM) components, resulting in skin laxity and wrinkles [22]. Although the puerarin solution does not significantly increase the content of type I collagen (COL-1), it increases the expression level of COL-1 from 0.263 to 1.861 and reduces the content from 9.933 to 5.783 ng/mL and expression level from 4.533 to 2.453 of matrix metalloproteinase 1 (MMP-1) (Figure 6), showing anti-aging effects.

### 3.4. Protective Mechanism of Puerarin against Oxidative Stress in Mice

#### 3.4.1. Blood Routine Test Results

It can be seen from Table A1 that when treating and UVA irradiating mice for two months, there are no significant differences in white blood cells, red blood cells, platelets, hemoglobin, neutrophils, and lymphocytes among the groups, and the measured values are within the normal range, showing that VC and puerarin have no side effects on the immune system of mice.

#### 3.4.2. Moisturizing Experiment

The moisture content (MMV) in the skin is crucial for maintaining its fullness and luster. If the skin moisture content drops, the skin becomes prone to dryness, roughness, and wrinkles, and transepidermal water loss (TEWL) is often used as an indicator of the health of the skin barrier [23]. After one month of feeding, sample loading, and UVA radiation, the MMV of the puerarin group increases to a certain extent from 41.82 to 44.79 C.U., while TEWL decreases significantly from 60.37 to 16.1 g/hm^2^ compared with the model group (Figure 7a,b), indicating that puerarin has a certain ability to moisturize and repair the skin barrier.

#### 3.4.3. Total Antioxidant Capacity and CAT Enzyme Activity in Mice Tissue

Puerarin significantly increased the total antioxidant capacity from 0.486 to 0.568 mmol/mg protein and CAT activity from 2.02 × 10^4^ to 2.478 × 10^4^ units/mg protein in mouse skin compared with the model group (Figure 7c,d).

#### 3.4.4. Skin Histological Observation

UVA irradiation caused noticeable wrinkles and much less skin tightness than in the control group (Figure 8d). The epidermis of the blank group (Figure 8a) was reasonably smooth, and the dermal collagen fibers were tightly arranged (Figure 8b,c), according to H&E-stained tissue sections (Figure 8a), but the epidermis of the model group (Figure 8e,f) displayed evident uneven hyperplasia and hyperkeratosis. Long-term exposure to UVA causes irregular thickening of the epidermis, an adaptive response of the skin to reduce UVA penetration that is also seen as a sign of skin damage [24]. In addition, the dermis of the model group had much fewer and fragmented collagen fiber bundles (Figure 8b). The degree of cuticle hyperplasia was less severe in the VC (Figure 8h,i)and RFP puerarin (Figure 8k,l) groups as compared to the model group, and the collagen arrangement in the dermis was more compact (Figure 8c,d). These findings imply that RFP puerarin can, to a considerable extent, mitigate UVA-induced skin photoaging.

## 4. Discussion

Long-term exposure to UVA radiation causes oxidative stress, which accelerates skin aging [25]. One of the most significant endogenous antioxidant stress protection mechanisms is the KEAP1-Nrf2/ARE pathway. When Nrf2, a critical transcription factor that regulates antioxidant response, dissociates from KEAP1, a 69 kDa protein that serves as a redox sensor and is negatively regulated with Nrf2, it translocates to the nucleus where it recognizes and binds ARE to initiate the transcription and expression of downstream antioxidant-related genes to exert antioxidant effect after forming a heterodimer with the small MAF protein [26]. Oxidative stress is also a mechanism of inflammation and tissue damage. Although Nrf2 has been shown to be an important mediator of NLRP3 and AIM2 inflammasome activation and also plays a promoting role in atherosclerosis (type II diabetes mellitus associated with chronic inflammation), it has also been shown that Nrf2 has a protective role in sepsis, asthma, infectious diseases, and other diseases [27]. Nrf2 may play different roles under different physiological conditions. However, the Nrf2/ARE signaling pathway has antioxidant and cytoprotective effects, which are essential for the body to resist inflammatory tissue damage. Nrf2-mediated upregulation of cytoprotective enzymes is closely related to cell resistance to infection and inflammation and the reduction of pro-inflammatory mediators. For example, HO-1 and NQO1, the representative stress response proteins in this pathway, have antioxidant and anti-inflammatory effects [28]. It has been widely reported that active antioxidant ingredients in plants can improve the endogenous antioxidant capacity to inhibit oxidative stress injury by activating the antioxidant signaling pathways. Functional potato bioactive peptide induces the decomposition of the KEAP1-Nrf2 complex by the AKT serine/threonine kinase signaling pathway in hypertensive rats, resulting in a higher expression level of endogenous antioxidant protein in the kidney to counteract renal damage [29]. Zerumbone, a natural sesquiterpenoid from Zingiber zerumbet rhizomes, facilitates the activation of Nrf2 to mediate the expression of antioxidant genes, thus showing anti-photoaging effects in UVA-irradiated HSF [30]. Honey/chamomile extracts inhibit KEAP1 activity and upregulate Nrf2 expression in keratinocytes [31]. Similarly, puerarin also has the effect of resisting oxidative stress through the Nrf2 pathway. It has potent antioxidant activity by activating Nrf2 to induce gene transcription expression related to glutathione biosynthesis and is considered a promising drug for the treatment of Parkinson’s disease [32]. Puerarin elevates the expression and protein content of Nrf2, as well as antioxidant enzymes such as CAT, GSH, SOD, HO-1, and NQO1, in mice with sodium glucan sulfate-induced colitis, indicating apparent oxidative protective effects [33]. Our findings are consistent with the above results that puerarin helps HSF to resist UVA radiation-induced oxidative stress by modulating the KEAP1-Nrf2/ARE signaling pathway. After treatment with 12 µg/mL of puerarin, the transcription and expression of KEAP1 are effectively reduced in UVA-irradiated HSF, and the nuclear transfer and transcription of Nrf2 are promoted (Figure 3). In addition, the activities and expressions of antioxidant enzymes downstream of the Nrf2 pathway are increased (Figure 4), which is also similar to many existing research results on puerarin. Treatment with puerarin increases the activity of enzymes in the antioxidant defense system, including SOD, CAT, GSH-Px, glutathione reductase, and glutathione s-transferase, while lowering ROS and MDA levels, thus assisting in the recovery of liver damage [34,35].

The three major enzymes in the antioxidant enzyme system (SOD, CAT, and GSH-Px) work together to decompose or transform ROS and lipid peroxides into harmless substances through diverse reactions, forming a complete antioxidant chain [36]. HO-1 degrades heme with potential pro-inflammatory and pro-oxidation effects into CO, biliverin, and Fe2+, while biliverin can effectively scavenge ROS, and CO hydrolysates generally have antioxidant effects [37,38]. NQO1 reduces quinones and their derivatives with NADH or NADPH as electron donors, preventing their further reaction to produce ROS, or protects cells from various metabolic oxidative stress responses by maintaining reduced forms of ubiquinones and α-tocopherol quinones [39]. GCLC contributes to glutathione production [40]. Nrf2 activation enhances the transcription and expression of the antioxidant enzymes, thereby improving the total antioxidant capacity of cells, reducing ROS production and the lipid peroxidation of cell membranes, and ultimately resisting cell damage caused by oxidative stress [41], which is consistent with the experimental results (Figure 2).

Moreover, ROS activates the mitogen-activated protein kinase (MAPK) family, which includes ERK, JNK, and P38 kinase [42]. The phosphorylated activation of ERK, JNK, and P38, independently or synergistically, activates the Nrf2 pathway by inducing Nrf2 phosphorylation and promoting its separation from KEAP1. Mingyi Zhao et al. [43] found that MAPK signaling is an upstream regulator of the Nrf2 pathway, and aminolevulinic acid, in combination with sodium ferrous citrate, activates the Nrf2/HO-1 signaling pathway by activating MAPK signaling, protecting cardiomyocytes from hypoxia-induced apoptosis. In mouse liver sinusoid endothelial cells, Gastrotin upregulates HO-1 and Nrf2 by phosphorylating P38 to alleviate H_2_O_2_-induced oxidative damage [44]. Butein has the potential to inhibit obesity-related metabolic syndrome by phosphorylating P38 to activate the Nrf2/HO-1 pathway [45]. MAPK activation causes the overexpression of the transcription factor activator protein 1, which leads to an increase in MMPs [46], a family of zinc-dependent enzymes responsible for ECM component degradation. MMP-1 is one of the major collagenases that initiate the degradation of Type I collagen, specifically cutting it into the specific *N*-terminus and *C*-terminus, which are further hydrolyzed by other MMPs [47]. Collagens, as the major components of the ECM of dermal connective tissue, play a vital part in the stability and tensile strength of the skin, and Type I collagen is one of the most important [48]. In UVB-irradiated HSF, vicenin-2 significantly decreases the phosphorylation of ERK, JNK, and P38, as well as AP-1 and MMPs protein levels [49]. Gelatin peptides from Pacific cod skin suppress ERK and P38 phosphorylation, lower AP-1 levels and inhibit the overexpression of MMPs caused by UV exposure [50]. Through the JNK/C-Jun/CYP7A1 pathway, puerarin efficiently decreases the phosphorylation level of JNK in mouse livers produced by carbon tetrachloride, contributing to the reduction of oxidative stress and hyperlipidemia [51]. Furthermore, puerarin helps to reduce cardiac hypertrophy induced by angiotensin II and retinal damage induced by the accumulation of excessive iron in the retina by inhibiting the excessive phosphorylation of ERK and P38 [52,53]. In this experiment, after puerarin treatment, the expression levels of JNK, ERK, and P38 mRNA in UVA-damaged cells are significantly decreased (Figure 5). MMP-1 mRNA expression level is dramatically reduced, as is MMP-1 content (Figure 6). We hypothesize that puerarin protects HSF against photoaging by lowering JNK, ERK, and P38 mRNA expression, inhibiting their phosphorylation and limiting MMP-1 production. As shown by Feng Chen et al. [54], UVB-mediated overexpression and phosphorylation of ERK1, JNK1, and P38 in HACAT cells are prevented by 6-Shogaol. However, one study has shown that puerarin does not regulate the phosphorylation of ERK1/2, JNK, and p38 in PC12 cells exposed to lead acetate, although it attenuated oxidative stress by regulating the expression of GCLC mRNA and protein levels, increasing glutathione levels, eliminating ROS, and reducing lipid peroxidation [55]. Therefore, the effects of puerarin on the MAPK pathway require further study.

## 5. Conclusions

The KEAP1-Nrf2/ARE antioxidant pathway is activated after puerarin treatment, and the content and expression of GSH-Px, SOD, CAT, HO-1, NQO1, and GCLC are upregulated, increasing the overall antioxidant ability of cells to remove ROS and MDA generated by UVA radiation. Meanwhile, puerarin, as an antioxidant, suppresses JNK, ERK, and P38 mRNA expression, which decreases MMP-1 content and expression, and increases COL-1 expression, resulting in a skin anti-aging effect. In addition, puerarin is hydrating and moisturizing on mouse skin, as well as showing a certain level of resistance to UVA oxidative stress. We propose that puerarin may provide an effective defense against UVA-induced skin photoaging.

## Figures and Tables

**Figure 1 nutrients-14-04724-f001:**
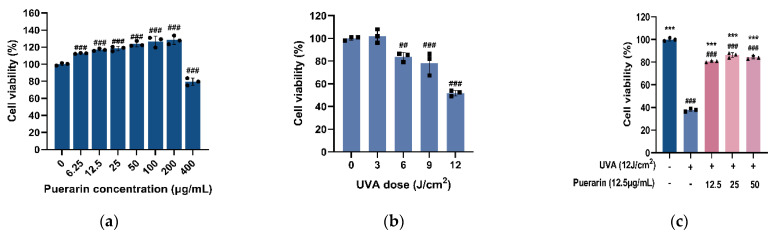
Effects of puerarin on UVA-induced cytotoxicity in human skin fibroblasts (HSF): (**a**) Toxicity of different concentrations of puerarin on HSF; (**b**) Toxicity of different doses of UVA on HSF; (**c**) Preventative effects of puerarin on cytotoxicity in UVA-irradiated HSF. ^##^ *p* < 0.01, ^###^ *p* < 0.001 as compared to the control group; *** *p* < 0.001 as compared to the damage model group; Values do not have a common mark (^#^, *) when *p* > 0.05, (*n* = 3).

**Figure 2 nutrients-14-04724-f002:**
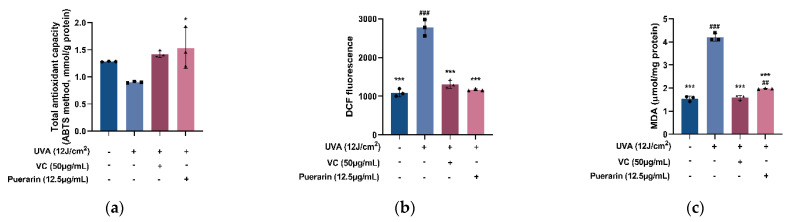
Effects of puerarin on cellular antioxidant indexes: (**a**) Effects of puerarin on total antioxidant capacity of HSF with UVA-induced oxidative damage. (**b**) Effects of puerarin on reactive oxygen species (ROS) production induced by UVA stimulation. (**c**) Effects of puerarin on lipid oxidation product malondialdehyde (MDA) content in HSF with UVA-induced oxidative damage. ^##^ *p* < 0.01, ^###^
*p* < 0.001 as compared to the control group; * *p* < 0.05, *** *p* < 0.001 as compared to the damage model group; Values do not have a common mark (^#^, *) when *p* > 0.05, (*n* = 3).

**Figure 3 nutrients-14-04724-f003:**
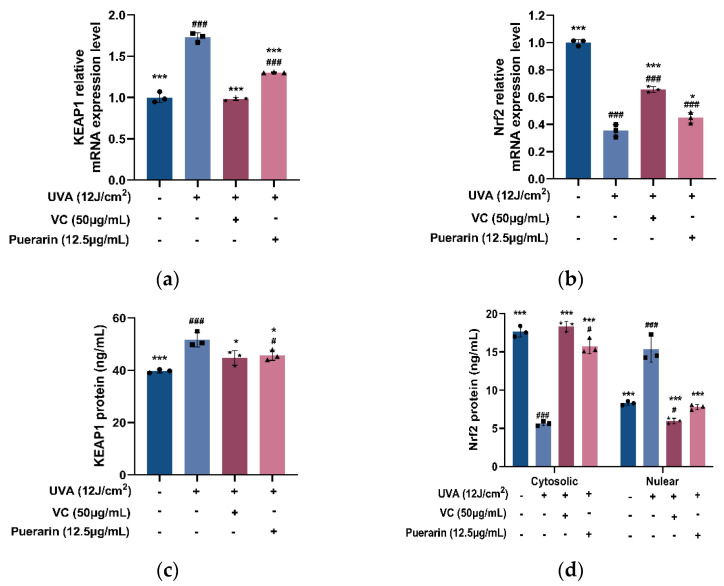
Effects of puerarin on UVA-induced Nrf2 signaling in HSF: (**a**,**b**) RT-PCR analysis of KEAP1 and NRF2 expression levels; (**c**,**d**) ELISA analysis of KEAP1 proteins and Nrf2 proteins in nucleus and cytosol. ^#^ *p* < 0.05, ^###^
*p* < 0.001 as compared to the control group; * *p* < 0.05, *** *p* < 0.001 as compared to the damage model group; Values do not have a common mark (^#^, *) when *p* > 0.05, (*n* = 3).

**Figure 4 nutrients-14-04724-f004:**
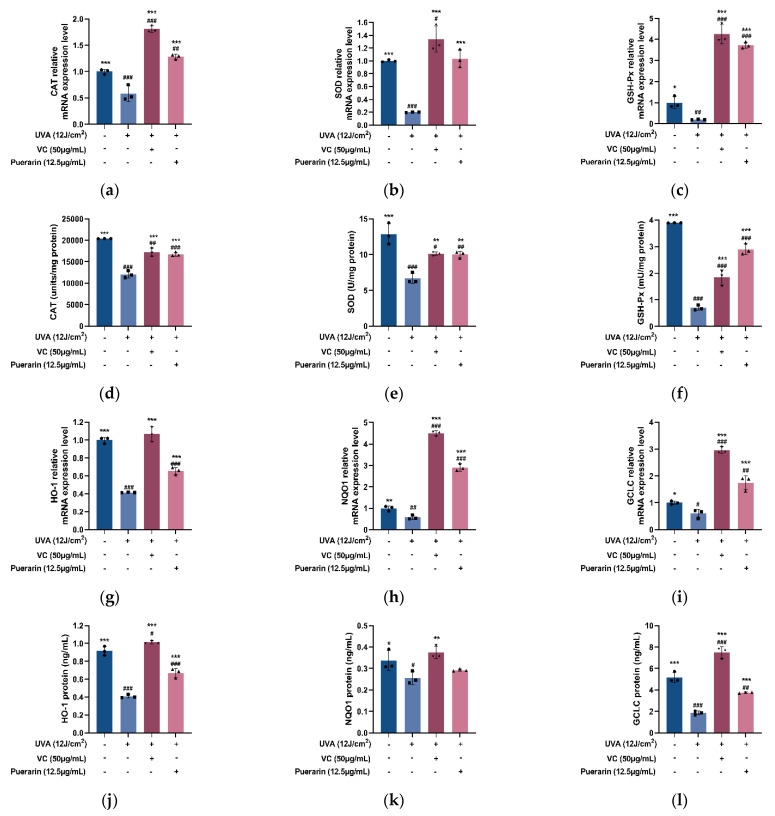
Effects of puerarin on UVA-induced antioxidase in HSF: (**d**–**f**) RT-PCR analysis of puerarin on UVA-induced SOD, CAT, and GSH-Px expression in HSF; (**a**–**c**) ELISA analysis of puerarin on UVA-induced SOD, CAT, and GSH-Px; (**j**–**l**) RT-PCR analysis of puerarin on UVA-induced HO-1, NQO1, and GCLC expression in HSF; (**g**–**i**) ELISA analysis of puerarin on UVA-induced HO-1, NQO1, and GCLC. ^#^ *p* < 0.05, ^##^ *p* < 0.01, ^###^
*p* < 0.001 as compared to the control group; * *p* < 0.05, ** *p* < 0.01, *** *p* < 0.001 as compared to the damage model group; Values do not have a common mark (^#^, *) when *p* > 0.05, (*n* = 3).

**Figure 5 nutrients-14-04724-f005:**
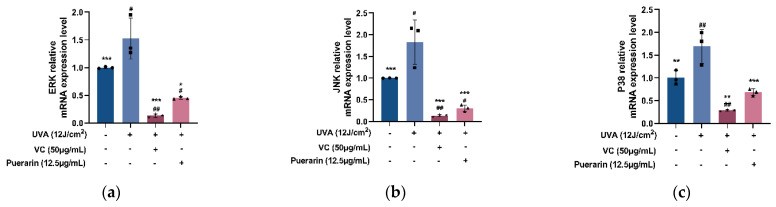
Effects of puerarin on UVA-induced mitogen-activated protein kinase (MAPK) signaling in HSF: (**a**–**c**) RT-PCR analysis of human extracellular signal-regulated kinase (ERK), human c-Jun amino-terminal kinase (JNK), and P38 mRNA expression levels after treatment and UVA irradiation of samples. ^#^ *p* < 0.05, ^##^ *p* < 0.01 as compared to the control group; * *p* < 0.05, ** *p* < 0.01, *** *p* < 0.001 as compared to the damage model group; Values do not have a common mark (^#^, *) when *p* > 0.05, (*n* = 3).

**Figure 6 nutrients-14-04724-f006:**
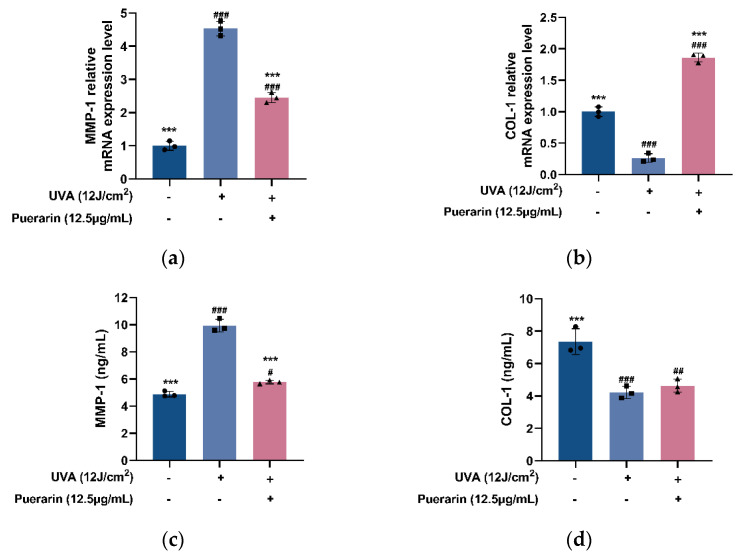
Effects of puerarin on UVA-induced skin aging markers in HSF: (**a**,**b**) Effects of puerarin on MMP-1 and COL-1 expression of HSF with UVA-induced oxidative damage; (**c**,**d**) Effects of puerarin on MMP-1 and COL-1 content of HSF with UVA-induced oxidative damage. ^#^ *p* < 0.05, ^##^ *p* < 0.01, ^###^
*p* < 0.001 as compared to the control group; *** *p* < 0.001 as compared to the damage model group; Values do not have a common mark (^#^, *) when *p* > 0.05, (*n* = 3).

**Figure 7 nutrients-14-04724-f007:**
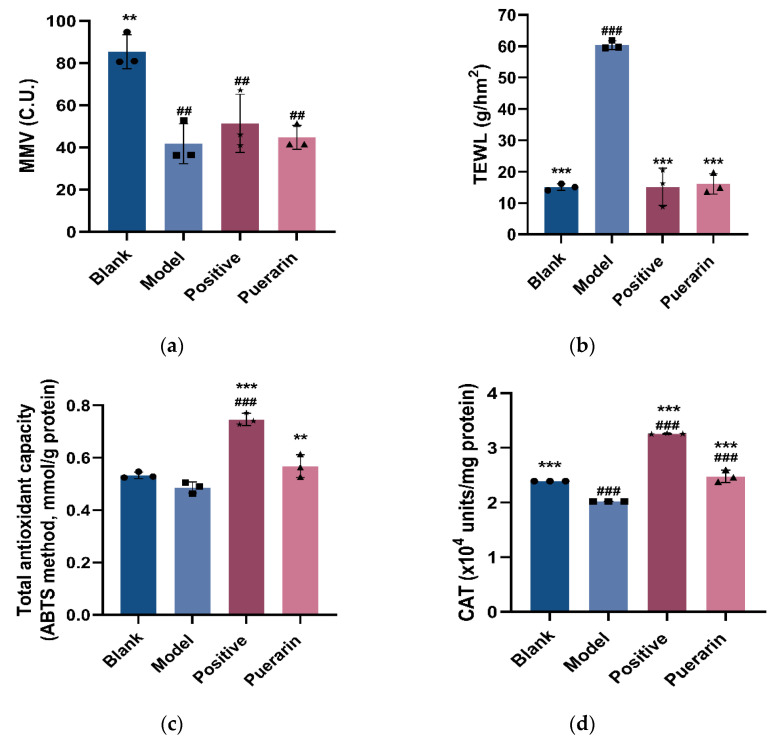
Effects of puerarin on UVA irradiation in mouse skin: (**a**) Results of moisture content (MMV) measurement value in mice skin; (**b**) Results of transepidermal water loss (TEWL) in mice skin; (**c**) Effects of puerarin on antioxidant capacity in mice after UVA irradiation; (**d**) Effects of puerarin on CAT activity in mice after UVA irradiation. ^##^ *p* < 0.01, ^###^
*p* < 0.001 as compared to the control group; ** *p* < 0.01, *** *p* < 0.001 as compared to the damage model group; Values do not have a common mark (^#^, *) when *p* > 0.05, (*n* = 3).

**Figure 8 nutrients-14-04724-f008:**
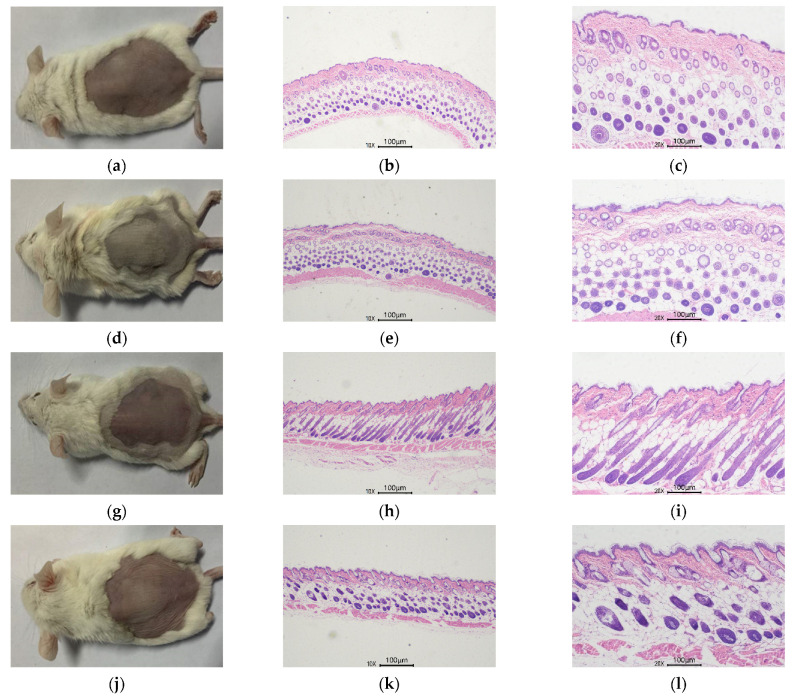
Effects of puerarin on mouse skin H&E staining: (**a**) Blank group mice; (**b**) Blank group skin H&E staining (10×); (**c**) Blank group skin H&E staining (20×); (**d**) Model group mice; (**e**) Model group skin H&E staining (10×); (**f**) Model group skin H&E staining (20×); (**g**) Positive control group mice; (**h**) Positive control group skin H&E staining (10×); (**i**) Positive control group skin H&E staining (20×); (**j**) Puerarin group mice; (**k**) Puerarin group skin H&E staining (10×); (**l**) Puerarin group skin H&E staining (20×).

**Table 1 nutrients-14-04724-t001:** RT-PCR Primer Sequences.

Gene	Direction	Primer Pair Sequence (5′→3′)
β-actin	F	TGGCACCCAGCACAATGAA
	R	CTAAGTCATAGTCCGCCTAGAAGCA
Nrf2	F	CAACTCAGCACCTTGTATC
	R	TTCTTAGTATCTGGCTTCTT
KEAP1	F	GGAGGCGGAGCCCGA
	R	GATGCCCTCAATGGACACCA
HO-1	F	CAAGCGCTATGTTCAGCGAC
	R	GCTTGAACTTGGTGGCACTG
NQO1	F	CAGCCAATCAGCGTTCGGTA
	R	CTTCATGGCGTAGTTGAATGATGTC
GCLC	F	CAGTCAAGGACCGGCACAAG
	R	CAAGAACATCGCCTCCATTCAG
ERK	F	TGTTCCCAAATGCTGACTCCAA
	R	TCGGGTCGTAATACTGCTCCAGATA
JNK	F	CTGTGTGGAATCAAGCACCTTCA
	R	CTGGCCAGACCGAAGTCAAGA
P38	F	TTAACAGGATGCCAAGCCATGA
	R	GGCACCAATAAATACATTCGCAAAG

## Data Availability

Not applicable.

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
