# Peer review of "Puerarin Reduces Oxidative Damage and Photoaging Caused by UVA Radiation in Human Fibroblasts by Regulating Nrf2 and MAPK Signaling Pathways"

_nutrients, 2022, doi:10.3390/nu14224724_

Round 1

Reviewer 1 Report

Concerns:

1.    Introduction: Kunming mouse model needs to be described and it’s use needs to be  justified.

2.    Method: The methods for cultured cells described very poorly and needs to be improved. For example:"2.3. Measurement of Cytotoxicity": CCK8  method and abbreviation need  to be described. Also, the purpose of Ref 17 is unclear. Reference 17 describes MTT kit to evaluate cytotoxicity, not the CCK8 method. The dose of  Puerarin in vitro  and in vivo needs to be justified. How Puerarin solution was made for administration in cultured cell and in vivo?

3.    Each mouse group needs to be described “four mice groups: model, blank, positive control VC, and puerarin”?

4.    Lane 106- “2.5. Antioxidant Effects of Puerarin on UVA-induced Oxidative Stress” is confusing, since authors measured vilabilty.Thus, Fig2A results  are confusing, too.

5.    Lane 146- “ being coated with the samples”-how the Puerarin  and other samples were prepared and administered?The description of the this part of methods needs to be improved.

6.    Most Figures quality can be improved.

7.    Authors indicated that they have Human Nuclear Factor E2-related Factor  (Nrf2) ELISA Kit for NRF analysis- how they demonstrated nuclear or cytosolic location of NRF2? Since oxidative stress activates NRF2 with consequent activation of its antioxidants enzymes targets,  it is unclear why UVA exposure leads to the opposite results in this study (Fig 3 B and D) and Fig4?

8.    Discussion: The underlying regulatory role of Nrf2 needs to be discussed in the light that Nrf2 also possesses pro-inflammatory effect, depending on its regulation and stability, resulting in the activation of NLRP3 as well as inflammation in macrophages (JBC,  VOL. 289, NO. 24, pp. 17020 –17029).

9.    Some references are not widely accessible, e.g. #13.

Author Response

Detailed response to reviewer 1’s comments

Oct. 26, 2022

Manuscript ID: nutrients-1965641

Title: Puerarin Reduces Oxidative Damage and Photoaging Caused by UVA

Radiation in Human Fibroblasts by Regulating Nrf2 and MAPK Signaling Pathways

Authors: Qiuting Mo, Shuping Li, Shiquan You, Dongdong Wang, Jiachan Zhang, Meng Li *, Changtao Wang

Received: 27 September 2022

E-mails: 2130041028@st.btbu.edu.cn, shupinglee@126.com, shiquan.you@starealife.com, wdd@btbu.edu.cn, 20120720@btbu.edu.cn, limeng@btbu.edu.cn, wangct@th.btbu.edu.cn

Dear referee,

Thank you very much for giving us this opportunity to revise our manuscript. The comments and suggestions made on our manuscript are very encouraging and helpful. Detailed point-by-point responses to all comments are provided in the following pages. Note that the comments are presented in italics, and our responses are in Roman and blue font. All changes made to the manuscript were marked using the “Track Changes” function. In addition, we addressed all these major points and other issues carefully and revised the manuscript accordingly. Please let me know if you have any further questions.

Sincerely,

Meng Li

Beijing Key Lab of Plant Resource Research and Development, College of Chemistry and Materials Engineering, Beijing Technology and Business University, Fucheng Road, Beijing 100048, China

Tel.: +86-13426015179

E-mail: limeng@btbu.edu.cn

  1. Introduction: Kunming mouse model needs to be described and it’s use needs to be justified.

Reply: Thanks for giving us the opportunity to revise the manuscript and your comments.

The mouse model was established by reference to Ref 17. The details have been rewritten and the approval number for animal testing is provided in the Institutional Review Board Statement.

“Twenty female mice weighing 18 ± 2 g were purchased from Beijing Vital River Laboratory Animal Technology Co., Ltd. Then they were divided into 4 groups randomly: blank group (deionized water), model group (deionized water + UVA), positive control group (50 μg/mL VC + UVA), puerarin group (12.5 μg/mL + UVA). Puerarin was predissolved with DMSO and then diluted with distilled water. The concentration of DMSO in puerarin solution was less than 0.1%. Each group was housed in a cage with a temper-ature of 22-25 ° C and a circadian cycle of 12 h of light and 12 h of darkness. They consumed food and liquids ad libitum. After one week of acclimation to the environment, the experiment was carried out. The same part of the back of each mouse was depilated with depilation cream, and the depilation area was 3 x 3 cm. After 1 h, the sample was applied at the dosage of 1 mL/d. After 30 min, they were irradiated with UVA at a dose of 50 J/cm2. The experimental period was 30 days [17].”

“Institutional Review Board Statement: All animal experiments were authorized by the Institutional Animal Care and Use Committee (IACUC) at Beijing Vital River Laboratory Animal Technology Co., Ltd. The ap-proval number was P2022015.”

  1. Method: The methods for cultured cells described very poorly and needs to be improved. For example:"2.3. Measurement of Cytotoxicity": CCK8 method and abbreviation need to be described. Also, the purpose of Ref 17 is unclear. Reference 17 describes MTT kit to evaluate cytotoxicity, not the CCK8 method. The dose of  Puerarin in vitro  and in vivo needs to be justified. How Puerarin solution was made for administration in cultured cell and in vivo?

Reply: Thanks for pointing out the questions.

The abbreviation CCK-8 comes from the Cell Counting Kit-8. And we have described the CCK-8 method in "2.3. Measurement of Cytotoxicity". Ref 17 was quoted for the purpose of referring to the time required for cell seeding and sample treatment.

Anyway, we are sorry for the confusion caused by the inappropriate reference. It has been changed for a more appropriate reference. The dosage of puerarin in cell and animal experiments was the same as 12.5 μg/mL, which was determined by cytotoxicity assay (Fig.1a). Because of the solubility of puerarin, it was predissolved with DMSO and then diluted with serum-free DMEM or distilled water, and the DMSO content of puerarin solution was less than 0.1%. Details of puerarin solution administration in cell and animal experiments have been described in the methods section.

“Cell Counting Kit-8 (CCK-8) kit.”

“After puerarin treatment for 24 h, it was changed to 100 µL DMEM containing 10% CCK8 solution for 2 h. The CCK-8 approach is based on 2-(2-Methoxy-4-nitrophenyl)-3-(4-nitrophenyl)-5-(2,4-disulfophenyl)-2H-tetrazoliumsodiumsalt (WST-8) which is an upgraded alternative to 3-(4,5)-dimethylthiahiazo(-z-y1)-3,5-di-phenytetrazolium-romide (MTT). Absorbance was determined at 450 nm.”

“Zhang, Y.; Wang, D.; Fu, H.; Zhao, D.; Zhang, J.; Li, M.; Wang, C. Protective effects of extracellular proteins of Saccharomycopsis fibuligera on UVA-damaged human skin fibroblasts. Journal of Functional Foods 2022, 88, doi:10.1016/j.jff.2021.104897.”

“HSF in good condition at the logarithmic growth stage were seeded on 96-well plates at a density of 8 × 103 cells/well for 12 h, then puerarin solution was added at concentrations of 0, 6.25, 12.5, 25, 50, 100, 200, and 400 µg/mL. Puerarin was predissolved with dimethyl sulfoxide (DMSO) followed by gradient dilution with DMEM without fetal bovine serum. The concentration of DMSO in 400 µg/mL puerarin solution was 0.1%. After puerarin treatment for 24 h [16], it was changed to 100 µL serum-free DMEM containing 10% CCK8 solution for 2 h.”

“Puerarin was predissolved with DMSO and then diluted with distilled water. The concentration of DMSO in puerarin solution was less than 0.1%. Each group was housed in a cage with a temper-ature of 22-25 ° C and a circadian cycle of 12 h of light and 12 h of darkness. They consumed food and liquids ad libitum. After one week of acclimation to the environment, the experiment was carried out. The same part of the back of each mouse was depilated with depilation cream, and the depilation area was 3 x 3 cm. After 1 h, the sample was applied at the dosage of 1 mL/d. After 30 min, they were irradiated with UVA at a dose of 50 J/cm2. The experimental period was 30 days [17].”

  1. Each mouse group needs to be described “four mice groups: model, blank, positive control VC, and puerarin”?

Reply: Thanks for your comments.

We have polished the description to make it easier for the reader to understand the grouping of animal experiments.

“Twenty female mice weighing 18 ± 2 g were purchased from Beijing Vital River Laboratory Animal Technology Co., Ltd. Then they were divided into 4 groups ran-domly: blank group (deionized water), model group (deionized water + UVA), positive control group (50 μg/mL VC + UVA), puerarin group (12.5 μg/mL + UVA).”

  1. Lane 106- “2.5. Antioxidant Effects of Puerarin on UVA-induced Oxidative Stress” is confusing, since authors measured vilabilty.Thus, Fig2A results are confusing, too.

Reply: Thanks for giving us the opportunity to revise the manuscript and pointing out the question.

We are sorry for the confusion caused by our poor choice of words. "2.5. protection Effects of Puerarin on UVA-induced Oxidative Stress" would better express our purpose.

“The protective effects of puerarin on oxidative stress were evaluated by detecting whether puerarin improves the viability of oxidation-stimulated cells.”

  1. Lane 146- “ being coated with the samples”-how the Puerarin and other samples were prepared and administered?The description of the this part of methods needs to be improved.

Reply: Thanks for pointing out the question.

We have refined the description of this part of the methods.

“Twenty female mice weighing 18 ± 2 g were purchased from Beijing Vital River Laboratory Animal Technology Co., Ltd. Then they were divided into 4 groups ran-domly: blank group (deionized water), model group (deionized water + UVA), positive control group (50 μg/mL VC + UVA), puerarin group (12.5 μg/mL + UVA). Puerarin was predissolved with DMSO and then diluted with distilled water. The concentration of DMSO in puerarin solution was less than 0.1%. Each group was housed in a cage with a temper-ature of 22-25 ° C and a circadian cycle of 12 h of light and 12 h of darkness. They consumed food and liquids ad libitum. After one week of acclimation to the environ-ment, the experiment was carried out. The same part of the back of each mouse was depilated with depilation cream, and the depilation area was 3 x 3 cm. After 1 h, the sample was applied at the dosage of 1 mL/d. After 30 min, they were irradiated with UVA at a dose of 50 J/cm2. The experimental period was 30 days [17].”

  1. Most Figures quality can be improved.

Reply: Thanks for your great suggestions which helped a lot for us to improve this manuscript. We have modified all the figures to present the data better.

  1. Authors indicated that they have Human Nuclear Factor E2-related Factor (Nrf2) ELISA Kit for NRF analysis- how they demonstrated nuclear or cytosolic location of NRF2? Since oxidative stress activates NRF2 with consequent activation of its antioxidants enzymes targets, it is unclear why UVA exposure leads to the opposite results in this study (Fig 3 B and D) and Fig4?

Reply: Thanks for your comments. The nuclear and cytoplasmic proteins were prepared according to the nuclear and cytoplasmic protein extraction kit purchased from Beyotime, China. As for the doubt that UVA exposure caused the opposite result, we have added a speculative interpretation in the results section.

“It is hypothesized that puerarin reduces the protein translation of KEAP1 by inhibiting KEAP1 transcription, thereby preventing the degradation of Nrf2 in the cytoplasm. The UVA radiation destroys the intracellular antioxidant system, and the nuclear translocation of Nrf2 is inhibited, leading to the decrease of intracellular Nrf2 protein content. The effect of puerarin restores the cellular antioxidant system and increases the nuclear Nrf2 protein content.”

  1. Discussion: The underlying regulatory role of Nrf2 needs to be discussed in the light that Nrf2 also possesses pro-inflammatory effect, depending on its regulation and stability, resulting in the activation of NLRP3 as well as inflammation in macrophages (JBC, VOL. 289, NO. 24, pp. 17020 –17029).

Reply: Thanks for your great suggestions which helped a lot for us to improve this manuscript. In accordance with your comments, the underlying regulatory role of Nrf2 has been discussed in the discussion section.

“Oxidative stress is also a mechanism of inflammation and tissue damage. Although Nrf2 has been shown to be an important mediator of NLRP3 and AIM2 inflammasome activation, and also plays a promoting role in atherosclerosis, type II diabetes mellitus associated with chronic inflammation, it has also been shown that Nrf2 has a protective role in sepsis, asthma, infectious diseases and other diseases [27]. Nrf2 may play dif-ferent roles under different physiological conditions. However, Nrf2/ ARE signaling pathway has antioxidant and cytoprotective effects, which are essential for the body to resist inflammatory tissue damage. Nrf2-mediated upregulation of cytoprotective en-zymes is closely related to cell resistance to infection and inflammation and reduction of proinflammatory mediators. For example, HO-1 and NQO1, the representative stress response proteins in this pathway, have antioxidant and anti-inflammatory effects [28].”

9.Some references are not widely accessible, e.g. #13.

Reply: Thanks for pointing out the questions. After searching the literature, we did not find any reference similar to Ref 13, so we had to delete it.

Reviewer 2 Report

1. Line 94 CO2 needs to be revised as "CO2"

2. How did authors dissolve puerarin?

3. Institutional animal protocol approval number should be provided?

4. line 152, authors should never use term killed, instead to use "euthanized"

5. The authors should represent all the figures with all the data points used for each figure in the bar graphs provided!

6. In figure 1B, why did the cell viability decreased at 50ug/mL for puerarin? hence it is advisable to show all the data points!!

7. what is positive control? where is the vehicle control?

8. Just gene expression is not sufficient, protein data needs to be added.

9. why only total Nrf2 data is shown, the authors should show cellular as well as nuclear Nrf2 protein data? as the study wholly depend on antioxidant activity.

10. Did authors establish upstream markers for Nrf2 activation? a proper use of siRNA data should have been established for Nrf2 story?

Author Response

Detailed response to reviewer 2’s comments

Oct. 26, 2022

Manuscript ID: nutrients-1965641

Title: Puerarin Reduces Oxidative Damage and Photoaging Caused by UVA

Radiation in Human Fibroblasts by Regulating Nrf2 and MAPK Signaling Pathways

Authors: Qiuting Mo, Shuping Li, Shiquan You, Dongdong Wang, Jiachan Zhang, Meng Li *, Changtao Wang

Received: 27 September 2022

E-mails: 2130041028@st.btbu.edu.cn, shupinglee@126.com, shiquan.you@starealife.com, wdd@btbu.edu.cn, 20120720@btbu.edu.cn, limeng@btbu.edu.cn, wangct@th.btbu.edu.cn

Dear referee,

Thank you very much for giving us this opportunity to revise our manuscript. The comments and suggestions made on our manuscript are very encouraging and helpful. Detailed point-by-point responses to all comments are provided in the following pages. Note that the comments are presented in italics, and our responses are in Roman and blue font. All changes made to the manuscript were marked using the “Track Changes” function. In addition, we addressed all these major points and other issues carefully and revised the manuscript accordingly. Please let me know if you have any further questions.

Sincerely,

Meng Li

Beijing Key Lab of Plant Resource Research and Development, College of Chemistry and Materials Engineering, Beijing Technology and Business University, Fucheng Road, Beijing 100048, China

Tel.: +86-13426015179

E-mail: limeng@btbu.edu.cn

  1. 1. Line 94 CO2 needs to be revised as "CO2".

Reply: Thanks for giving us the opportunity to revise the manuscript and pointing out the question. We have revised it.

“HSF were grown in a DMEM medium supplemented with 10% fetal bovine serum and 1% Penicillin-Streptomycin Solution. The incubator was humidified with a temperature of 37°C and CO2 content of 5%. Trypsin was used for digestion and passage when the cell density was around 85%.”

  1. How did authors dissolve puerarin?

Reply: Thanks for your comments.

Because of the solubility of puerarin, it was predissolved with DMSO and then diluted with serum-free DMEM or distilled water, and the DMSO content of puerarin solution was less than 0.1%. Details of puerarin solution administration in cell and animal experiments have been described in the methods section.

“Puerarin was predissolved with dimethyl sulfoxide (DMSO) followed by gradient dilution with DMEM without fetal bovine serum. The concentration of DMSO in 400 µg/mL puerarin solution was 0.1%. After puerarin treatment for 24 h [16], it was changed to 100 µL serum-free DMEM containing 10% CCK8 solution for 2 h.”

“Puerarin was predissolved with DMSO and then diluted with distilled water. The concentration of DMSO in puerarin solution was less than 0.1%. Each group was housed in a cage with a temperature of 22-25 ° C and a circadian cycle of 12 h of light and 12 h of darkness. They consumed food and liquids ad libitum. After one week of acclimation to the environment, the experiment was carried out. The same part of the back of each mouse was depilated with depilation cream, and the depilation area was 3 x 3 cm. After 1 h, the sample was applied at the dosage of 1 mL/d. After 30 min, they were irradiated with UVA at a dose of 50 J/cm2. The experimental period was 30 days [17].”

  1. Institutional animal protocol approval number should be provided?

Reply: Thanks for your comments.

The approval number for animal testing is provided in the Institutional Review Board Statement section.

“Institutional Review Board Statement: All animal experiments were authorized by the Institutional Animal Care and Use Committee (IACUC) at Beijing Vital River Laboratory Animal Technology Co., Ltd. The approval number was P2022015.”

  1. line 152, authors should never use term killed, instead to use "euthanized".

Reply: Thanks for pointing out the question. We have revised it.

“After two months of feeding and before the mice were euthanized, the water content and transepidermal water loss of the back skin were measured using a water content tester (Corneometer® CM825, Courage+Khazaka Co., Germany) and transepidermal water loss tester (Tewameter® TM300, Courage+Khazaka Co., Germany) (Temperature: 27 ± 1℃, humidity: 40 ± 5%).”

  1. The authors should represent all the figures with all the data points used for each figure in the bar graphs provided!

Reply: Thanks for your great suggestions which helped a lot for us to improve this manuscript.

We have used all the data points of each figure to represent the figures according to your comments.

“The cell viability is 97.92% when serum free DMEM containing 0.1%DMSO was used for cytotoxicity test, indicating that DMSO had a negligible effect on cells. When the concentration of puerarin solution is 6.25–200 µg/mL, the survival rate of HSF is higher than 80%, and it even shows a proliferation effect on the cells which is proportional to the concentration. The cell viability rates are 112.67%, 116.92%, 118.29%, 123.55%, 126.51% and 128.26%, respectively. However, when the concentration is higher than 400µg/mL, the cell viability decreased to 79.39% which may have certain toxicity to HSF (Fig.1a). Low dose (3 and 6 J/cm2) UVA irradiation show 102% and 83.76% cell viability having no significant effect on cells (Fig. 1b). When the UVA irradiation is 9 J/cm2, the cell viability is 78.40%, but the effect on cell viability is still relatively small. When the UVA irradiation is 12 J/cm2, the cell viability is decreased to 51.74% (P < 0.001). Excessive exposure to UVA would cause irreversible damage to cells, so 12 J/cm2 irradiation was used to establish the oxidation model. Cell viability is significantly decreased after UVA stimulation (P < 0.001), but significantly increase to 80.76%, 86.32% and 84.43% in order after treatment with 12.5, 25, and 50 µg/mL puerarin solution, indicating protection from oxidative stress by UVA radiation (P < 0.001) (Fig. 1c). After comprehensive consideration, 12.5 µg/mL of puerarin was finally selected for subsequent experiments.”

“Compared with the model group (0.903 mmol/g protein), the total antioxidant capacity of puerarin group and positive control group are improved to 1.46 and 1.39 mmol/g protein, which slowed down the oxidative damage caused by UVA to cells (Fig.2a). Reactive oxygen species (ROS) and lipid oxidation product malondialdehyde (MDA) are the signature products of cellular oxidative stress [19,20]. UVA radiation causes a significant increase in ROS and MDA content (P < 0.001) (Fig.2b). However, 12.5 µg/mL of puerarin solution dramatically minimizes the generation of ROS (relative fluorescent value from 2774 to 1167) and MDA (from 4.2 to 1.98 µmol/mg protein) as a result of UVA damage (P < 0.001) (Fig.2c).”

“The Puerarin solution reduces the relative mRNA transcription of KEAP1 from 1.743 to 1.302 (Fig.3a) and the protein content from 51.68 to 45.77 ng/mL (Fig.3c). Puerarin promotes the relative mRNA transcription of Nrf2 from 0.353 to 0.448 (Fig.3b). By measuring the relative protein content of Nrf2 in cytoplasm and nucleus, it was found that after UVA treatment, the protein content of Nrf2 in nucleus decreases to 5.62 ng/mL, and the cytoplasmic protein increases to 15.36 ng/mL, while after puerarin treatment, the nuclear Nrf2 content is reversed to 15.76 ng/mL, and the cytoplasmic Nrf2 content is 7.797 ng/mL. It is hypothesized that puerarin reduces the protein translation of KEAP1 by inhibiting KEAP1 transcription, thereby preventing the degradation of Nrf2 in the cytoplasm.  The UVA radiation destroys the intracellular antioxidant system, and the nuclear translocation of Nrf2 is inhibited, leading to the decrease of intracellular Nrf2 protein content. The effect of puerarin restores the cellular antioxidant system and increases the nuclear Nrf2 protein content.”

“Activated Nrf2 induces the expression of antioxidant enzymes related to cell protection, thereby improving the resistance of cells to oxidative damage. In the experiment of UVA-stimulated HSF, UVA radiation significantly lowers the enzymes activity and mRNA expression of glutathione peroxidase (GSH-Px), CAT, SOD, HO-1, NQO1, and glutamate-cysteine ligase catalytic subunit (GCLC) (P < 0.01). However, puerarin has significant promoting effects (Fig.4). The relative transcript levels of antioxidant enzymes reduced by UVA irradiation in the inner cell are enhanced from 0.199 to 1.036, from 0.582 to 1.282, from 0.202 to 3.726, from 0.415 to 0.655, from 0.577 to 2.911, and from 0.597 to 1.741, respectively. The corresponding antioxidant enzyme activity also is in-creased sequentially from 12,070 to 16,757 units/mg protein, from 6.665 to 10 U/mg protein, from 0.701 to 2.899 mU/mg protein, from 0.409 to 0.67 ng/mL, from 0.256 to 0.292 ng/mL, and from 1.872 to 3.752 ng/mL.”

“RT-PCR results show that 12 µg/mL of puerarin solution significantly inhibits the mRNA expression levels of the human c-Jun amino-terminal kinase (JNK) from 1.826 to 0.306, human extracellular signal-regulated kinase (ERK) from 1.526 to 0.454, and P38 from 1.693 to 0.688 upregulated by UVA irradiation compared with the model group (P < 0.01) (Fig.5).”

“UVA-induced oxidative stress upregulates the expression of matrix metalloproteinases (MMPs), leading to skin collagen degradation and other extracellular matrices (ECM) components, resulting in skin laxity and wrinkles [22]. Although the puerarin solution does not significantly increase the content of type I collagen (COL-1), it increases the expression level of COL-1 from 0.263 to 1.861 and reduces the content from 9.933 to 5.783 ng/mL and expression level from 4.533 to 2.453 of matrix metalloproteinase 1 (MMP-1) (Fig.6), showing anti-aging effects.”

“The moisture content (MMV) in the skin is crucial for maintaining its fullness and luster. If the skin moisture content drops, the skin becomes prone to dryness, roughness and wrinkles, and transepidermal water loss (TEWL) is often used as an indicator of the health of the skin barrier [23]. After one month of feeding, sample loading, and UVA radiation, the MMV of the puerarin group increases to a certain extent from 41.82 to 44.79 C.U., while TEWL decreases significantly from 60.37 to 16.1 g/hm2 compared with the model group (Fig.7(a-b)), indicating that puerarin has a certain ability to moisturize and repair the skin barrier.”

“Puerarin significantly increased the total antioxidant capacity from 0.486 to 0.568 mmol/mg protein and CAT activity from 2.02×104 to 2.478×104 units/mg protein in mouse skin compared with model group (Fig.7 (c-d)).”

  1. In figure 1B, why did the cell viability decreased at 50ug/mL for puerarin? Hence it is advisable to show all the data points!

Reply: Thanks for your great suggestions.

The previous results found that when the concentration of puerarin was 50 ug/mL, the cell viability decreased probably because there was a certain difference in the number of cells when the cells were spread in the 96-well plate. However, the overall trend showed that puerarin promoted cell proliferation. We have repeated the experiment and found that puerarin solution at concentrations ranging from 6.25 to 200 µg/mL had a proliferative effect on cells, which was proportional to the concentration.

  1. what is positive control? where is the vehicle control?

Reply: Thanks for your comments. Referring to Ref 18, 50 µg/mL vitamin C (VC) was used as a positive control for antioxidant test. Because the concentration of DMSO in solution was less than 0.1%, the effect on cells and animals was negligible, therefore, we did not set a separate solvent control. The CCK8 assay was performed with serum-free DMEM containing 0.1%DMSO, and the cell viability was 97.92%, which confirmed the nontoxicity to cells.

  1. Just gene expression is not sufficient, protein data needs to be added.

Reply: Thanks for your great suggestions. However, we are sorry that due to time constraints and laboratory conditions, we cannot complete the detection of ERK, JNK and P38 phosphorylated proteins for the time being.

  1. why only total Nrf2 data is shown, the authors should show cellular as well as nuclear Nrf2 protein data? as the study wholly depend on antioxidant activity.

Reply: Thanks for your comments. The nuclear and cytoplasmic proteins were prepared according to the nuclear and cytoplasmic protein extraction kit purchased from Beyotime, China. Fig. 3d shows the protein content data of Nrf2 in cytoplasm and nucleus.

  1. Did authors establish upstream markers for Nrf2 activation? a proper use of siRNA data should have been established for Nrf2 story?

Reply: Thanks for giving us the opportunity to revise the manuscript and your comments.

Since we measured the content of Nrf2 protein in cytoplasm and nucleus, it was found that the nuclear displacement ratio of Nrf2 increased, indicating that the Nrf2 pathway was activated. And it was also found that the transcription and activity of antioxidant enzymes downstream of this pathway were significantly increased. Therefore, it is not necessary to establish the upstream activation marker of Nrf2 or perform transfection experiments.

Round 2

Reviewer 1 Report

N/A

Author Response

Detailed response to reviewer 1’s comments

Nov. 04, 2022

Manuscript ID: nutrients-1965641

Title: Puerarin Reduces Oxidative Damage and Photoaging Caused by UVA

Radiation in Human Fibroblasts by Regulating Nrf2 and MAPK Signaling Pathways

Authors: Qiuting Mo, Shuping Li, Shiquan You, Dongdong Wang, Jiachan Zhang, Meng Li *, Changtao Wang

Received: 27 September 2022

E-mails: 2130041028@st.btbu.edu.cn; shupinglee@126.com; 13269182262@163.com; wdd@btbu.edu.cn; xiaochan8787@163.com; limeng@btbu.edu.cn; wangct@th.btbu.edu.cn

Dear referee,

We are appreciated for your time on reviewing the manuscript. The comments and suggestions made on our manuscript are very encouraging and helpful.

Sincerely,

Meng Li

Beijing Key Lab of Plant Resource Research and Development, College of Chemistry and Materials Engineering, Beijing Technology and Business University, Fucheng Road, Beijing 100048, China

Tel.: +86-13426015179

E-mail: limeng@btbu.edu.cn

Reviewer 2 Report

Minor revisions need to be addressed

1.  ELISA 2.8, line 137. The authors have to reframe the sentence "Then the protein content of KEAP1, NQO1, HO-1, GCLC, COL-1 were detected",...... it needs be said as " Total protein was estimated from the cell lysates for KEAP1, NQO1, HO-1, GCLC, COL-1 assay."

2. Table 1. please provide the ref for primer sequences 

3. in all of the figure legends, provide how many replicates for each assay and how many times the assay was repeated like (n=3)??

4. The individual data points to be represented in the graph, when plotting the bar graph. 

5. each of the figure check the accuracy of the error plots, some error bars have both, some only have above bars? need to be consistent throughout?

Author Response

Detailed response to reviewer 2’s comments

Nov. 04, 2022

Manuscript ID: nutrients-1965641

Title: Puerarin Reduces Oxidative Damage and Photoaging Caused by UVA

Radiation in Human Fibroblasts by Regulating Nrf2 and MAPK Signaling Pathways

Authors: Qiuting Mo, Shuping Li, Shiquan You, Dongdong Wang, Jiachan Zhang, Meng Li *, Changtao Wang

Received: 27 September 2022

E-mails: 2130041028@st.btbu.edu.cn; shupinglee@126.com; 13269182262@163.com; wdd@btbu.edu.cn; xiaochan8787@163.com; limeng@btbu.edu.cn; wangct@th.btbu.edu.cn

Dear referee,

Thank you very much for giving us this opportunity to revise our manuscript. The comments and suggestions made on our manuscript are very encouraging and helpful. Detailed point-by-point responses to all comments are provided in the following pages. Note that the comments are presented in italics, and our responses are in Roman and blue font. All changes made to the manuscript were marked using the “Track Changes” function. In addition, we addressed all these major points and other issues carefully and revised the manuscript accordingly. Please let me know if you have any further questions.

Sincerely,

Meng Li

Beijing Key Lab of Plant Resource Research and Development, College of Chemistry and Materials Engineering, Beijing Technology and Business University, Fucheng Road, Beijing 100048, China

Tel.: +86-13426015179

E-mail: limeng@btbu.edu.cn

  1. ELISA 2.8, line 137. The authors have to reframe the sentence "Then the protein content of KEAP1, NQO1, HO-1, GCLC, COL-1 were detected",...... it needs be said as " Total protein was estimated from the cell lysates for KEAP1, NQO1, HO-1, GCLC, COL-1 assay."

Reply: Thanks for giving us the opportunity to revise the manuscript and pointing out the question. We have revised it.

“The cells were plated in 25cm2 flask at density of 5 × 106 cells/flask or 75cm2 flask at density of 1.5 × 107 cells/flask for 12 h. After sample processing and UVA irradiation, the nuclear and cytoplasmic proteins were prepared according to nuclear and cytoplasmic protein extraction kit. Then, the protein of Nrf2 in nuclear and cytoplasmic were detected. The cell lysates were prepared by cell lysis buffer. Total protein was estimated from the cell lysates for KEAP1, NQO1, HO-1, GCLC, COL-1 assay. Cell Culture Supernates were used to determine the protein content of MMP-1.”

  1. Table 1. please provide the ref for primer sequences

Reply: Thanks for your comments.

The Primer sequences were designed using PrimerExpress software based on the sequence genes published by the National Center for Biotechnology Information (NCBI), so there is no specific reference.

“The primers were designed using PrimerExpress software based on the gene sequences published in National Center for Biotechnology Information (NCBI), and the reference gene was β-actin.”

  1. in all of the figure legends, provide how many replicates for each assay and how many times the assay was repeated like (n=3)?

Reply: Thanks for your great suggestions which helped a lot for us to improve this manuscript.

According to your comments, we have provided the number (n = 3) of repetitions per trial in all the legends. In addition, the biological and technical repetitions of the experiments have been explained in the statistical analysis section.

“Each experiment included at least three biological repeats with three technical replicates.”

  1. The individual data points to be represented in the graph, when plotting the bar graph.

Reply: Thanks for pointing out the question. We have used all the data points of each figure to represent the figures according to your comments.

  1. each of the figure check the accuracy of the error plots, some error bars have both, some only have above bars? need to be consistent throughout?

Reply: Thanks for pointing out the question. We have modified all the figures to make them consistent in format.
